# Tying Together Multiscale Calculations for Charge Transport in P3HT: Structural Descriptors, Morphology, and Tie-Chains

**DOI:** 10.3390/polym10121358

**Published:** 2018-12-07

**Authors:** Evan D. Miller, Matthew L. Jones, Eric Jankowski

**Affiliations:** Micron School of Materials Science and Engineering, Boise State University, Boise, ID 83705, USA; evanmiller326@boisestate.edu (E.D.M.); mattyjones@boisestate.edu (M.L.J.)

**Keywords:** organic photovoltaics, charge transport, semi-empirical, kinetic Monte Carlo

## Abstract

Evaluating new, promising organic molecules to make next-generation organic optoelectronic devices necessitates the evaluation of charge carrier transport performance through the semi-conducting medium. In this work, we utilize quantum chemical calculations (QCC) and kinetic Monte Carlo (KMC) simulations to predict the zero-field hole mobilities of ∼100 morphologies of the benchmark polymer poly(3-hexylthiophene), with varying simulation volume, structural order, and chain-length polydispersity. Morphologies with monodisperse chains were generated previously using an optimized molecular dynamics force-field and represent a spectrum of nanostructured order. We discover that a combined consideration of backbone clustering and system-wide disorder arising from side-chain conformations are correlated with hole mobility. Furthermore, we show that strongly interconnected thiophene backbones are required for efficient charge transport. This definitively shows the role “tie-chains” play in enabling mobile charges in P3HT. By marrying QCC and KMC over multiple length- and time-scales, we demonstrate that it is now possible to routinely probe the relationship between molecular nanostructure and device performance.

## 1. Introduction

Organic photovoltaics (OPVs)—solar panels built with carbon-based compounds—are advantageous due to low-cost, scalable manufacturing methods [1,2]. OPVs have the potential to inexpensively meet growing energy demand with energy pay-back times in as little as one day with estimated 15% photoconversion efficiencies [3], a goal that has recently been surpassed by laboratory tandem devices [4]. The challenge in achieving mass-produced devices with similar efficiencies is controlling the spontaneous formation of nanostructures with thermodynamic self-assembly [5,6]. Obtaining favorable morphologies is complicated because the choice of ingredients, solvents [7], annealing methods [8,9,10], and processing temperatures [11] all affect self-assembly. Furthermore, it is challenging to predict which morphologies will exhibit the most favorable charge transport properties. To engineer better OPVs and ameliorate global climate change it is necessary to answer (1) “Which nanostructures are required for high device efficiency?”, and (2) “What processing protocols are required to obtain these structures?” In this article, we address the first question by identifying structure-performance relationships for the benchmark donor material poly-(3-hexylthiophene) (P3HT). The second issue is investigated for P3HT in a companion work [12].

In order to successfully evaluate structure-performance relationship, we require a performance descriptor that can quantifiably describe morphologies. One such property is the charge-carrier mobility, μ, which describes the speed that electrons and holes conduct through the semiconducting layer. Generally speaking, OPVs with faster carrier mobilities exhibit faster response times and better overall performance (although other properties such as optical absorption also ultimately affect performance) [1]. However, a key advantage of selecting the hole mobility as the performance metric is that it is widely applicable to all organic electronic devices—such as transistors and diodes. In this investigation, we therefore measure the hole mobility of P3HT in the absence of an electric field, $\mu_{0}$, which is similar to time-of-flight mobilities that are measured experimentally, and allows us to explore charge movement free of any electric field [13].

P3HT is the benchmark donor material in organic electronics, largely due to reasonable hole mobilities, ideal energy level placement and bandgap, excellent optical absorption, and good solution processability [14]. It is also widely studied experimentally—especially in the context of P3HT:fullerene OPVs where device power conversion efficiencies as high as 6.5% have been obtained [15]. In neat P3HT field effect transistors, high mobilities of 0.1–0.4 cm${}^{2}$/Vs have been measured for devices which contained microcrystalline domains surrounded by an amorphous matrix [16,17]. Time-of-flight mobilities for P3HT tend to be lower, $\mu =1\times 10^{-5}$ to $1\times 10^{-3}$ cm${}^{2}$/Vs, due to the lower charge density and absence of an electric field to drive the movement of charges [18,19,20,21]. These experiments have made excellent progress in linking the nanoscale polymer structure to device performance. For instance, high regioregularity [18,19,21] (i.e., a large proportion of the monomers in each chain have a consistent placement of the aliphatic side chain attached to the thiophene ring (see Figure 1a) and shorter polymer chains are both expected to result in advantageous molecular packing resulting in a high degree of crystalline order [20]. However, comprehensive experimental investigation of the relationship between morphology and charge motion is prohibited by time, expense, and difficulty.

Computational modeling provides insight into morphology and mobility that is inaccessible in experiments, and can more efficiently be used to evaluate how changes to processing parameters (temperature, solvent quality) tune performance. Techniques such as drift-diffusion [22,23], master equation [24,25], and kinetic Monte Carlo (KMC) [26,27,28,29] have all been successfully employed to investigate charge transport of model OPV morphologies. Drift-diffusion and lattice-based “mesoscale” KMC [30] can investigate device performance properties, but lose important details at molecular length-scales. Conversely, master equation and “molecular” KMC maintain the molecular resolution but require approximations such as periodic boundary conditions to investigate charge motion over distances relevant for devices [25,28,29]. Such methods have been used to investigate time-of-flight mobilities, some reporting values a few orders of magnitude higher than expected (1× 10${}^{0}$ to 1× 10${}^{3}$ cm${}^{2}$/Vs) [28,31,32], and others focusing on transfer integrals and inferred mobility without predicting mobility values [26,33]. For this investigation, we implement molecular KMC simulations, which are more computationally expensive than master equation techniques, but offer explicit spatial resolution of charges within the morphology [28,29]. Molecular KMC uses the positions, orientations, and energetics of electronically active portions of the molecules (chromophores), to determine the rates at which carriers can perform quantized “hops” between chromophore pairs. The hopping rates between every pair of chromophores in the system can be calculated in order to predict the expected motion of carriers through the system and the overall $\mu_{0}$ (subscript “0” signifies that there is no electric field, similar to time-of-flight experiments).

In this work we utilize morphologies generated in previous work using a model that provides state-of-the-art prediction accuracy validated by experiments, while still providing sufficient computational efficiency to facilitate the investigation of a large number of processing parameter combinations [12]. With these morphologies, we perform semi-empirical QCCs to obtain the chromophore energy levels and molecular KMC simulations to obtain hole mobilities for pristine P3HT thin films at ∼100 different state-points. We find that the structural order parameter developed previously, ψ, does not satisfactorily predict the observed charge carrier mobility in the thin films. Modifying the descriptor by including the variation of aliphatic bond lengths as a proxy for system-wide disorder, $\psi^{\prime }$, provides much better quantitative agreement between order parameter and charge mobility for the small “parameter sweep” systems. That said, we show that $\psi^{\prime }$ is less predictive of mobility for larger monodisperse systems (10× the number of chains) at experimentally interesting state-points. We propose that this is due to $\psi^{\prime }$ not taking into account the difficulty in charges hopping between crystallite grains of different orientations, effectively trapping carriers in the ordered crystallites. We therefore investigate the effect of a polydisperse distribution of chain lengths on mobility. We show that the longest chains in these distributions can span multiple clusters as “tie-chains” and mitigate the carrier trapping within crystals, bringing mobilities back in line with the predictions offered by $\psi^{\prime }$. This allows us to predict the processing conditions that result in the highest performing devices. Our finding that highly ordered structures may have low mobilities if connecting paths between ordered domains are absent paves the way for new analytical techniques to help link structure to device performance [31].

This article is structured as follows: in Section 2 we cover the important details of previous molecular dynamics (MD) work and discuss the methodology in using the MorphCT simulation package [34] to conduct KMC simulations and obtain zero-field mobilities. In Section 3.1 we then explore charge transport properties for a large set of morphologies generated using an optimized molecular model, and in Section 3.2 identify structural features important for charge transport in larger systems.

## 2. Methods

### 2.1. Molecular Dynamics Simulations

The P3HT morphologies studied here were previously predicted using MD simulations, and so only salient information will be covered here [12]. We investigate ∼100 morphologies generated from simulations using an adapted Optimized Performance for Liquid Simulations—United Atom forcefield to govern the non-bonded pair interactions (see Figure 1b). The united-atom model consists of three species of simulation beads: sulfur-S, aliphatic carbons-CT, and aromatic carbons-CA. The bonded constraints (bonds, angles, and dihedrals) are derived from the atomistic works of Huang and Bhatta [35,36], with modifications made to account for the reduced number of distinct atomic species utilized in the united-atom model. In the interest of computational performance, we make several additional modeling choices: solvent and electrostatic interactions are considered implicitly and thiophene rings are considered as rigid bodies (bonded constraints are held constant). The optimized model is shown to have excellent agreement with experimental results while accessing sufficient computational efficiency to explore a wide parameter space, with each simulation reaching equilibrium in tenable wall-clock simulation times [12].

In our previous work, we simulate cubic simulation volumes of side 7 nm containing 100 × 15 oligomer long chains at a variety of temperatures *T*, densities ρ, and implicit solvent solubility $\varepsilon_{s}$. $\varepsilon_{s}$ is a reciprocal measure of solubility, with low values corresponding to solvents that easily solvate P3HT. Here, all systems are at experimentally measured thin-film density, ρ = 1.11 g/cm${}^{3}$. However, we still consider a wide range of temperatures (80≤T≤1300 K in steps of 80 K) and solvents ($0.2\leq \varepsilon_{s}\leq 1.2$ in steps of 0.2). We find that the highest degrees of order are observed in a band located from low temperatures and “good” solvents (T∼ 250 K, $\varepsilon_{s}$ = 0.2) to high temperatures and “poor” solvents (T∼ 750 K, $\varepsilon_{s}$ = 1.2). We note that the *T* defined here corresponds only to the temperatures of the MD simulation, which affects the energetic disorder of the polymer chains but not subsequent charge transport calculations. We also curate larger systems (cubic simulation volumes of side 15 nm containing 1000 15-mers—chains with degree of polymerization 15) with differing degrees of order (as quantified by the order parameter, ψ): “amorphous”, “semi-crystalline”, and “crystalline” by terminating the evolution of an experimentally relevant state-point after different degrees of equilibration. This ensures that different degrees of structural order can be obtained without changing the energetic disorder arising from thermal vibrations, allowing us to divorce the effects of structure and energetic disorder on the charge transport properties.

To quantify the structural order of these systems, we develop an order parameter, ψ, which is defined as the fraction of thiophene rings comprising “large” clusters containing more than six thiophene rings out of the total number of thiophene rings in the system—a measure of the proportion of crystallinity. Two thiophenes are considered clustered if their centroids are within 6.6 Å, and if the normal vectors of the thiophene rings are oriented within 20${}^{\circ }$ of each other. These criteria were selected from a combination of the observed radial distribution functions, and other works which have shown such values produce a high degree of molecular orbital overlap between rings. A key aim of this work is to ascertain the efficacy of using ψ—a purely structural property of the morphology—to predict the charge transport of an arbitrary morphology.

### 2.2. Kinetic Monte Carlo Simulations

The charge transport calculations are performed using the MorphCT software package, running on Intel Xeon central processing units [29,34]. Mapping the structure to the mobility requires several processes operating over multiple length-scales. These are combined into an automated simulation pipeline, permitting us to analyse the molecular structure on Ångström length-scales and femtosecond time-scales, to carrier motion over hundreds of nanometers and microseconds. Firstly, the morphology is returned to the atomistic from the united-atom representation. For more strongly coarse-grained systems than those explored here, the interested reader is referred to the fine-graining methodology described in ref [29]. In the case of the united-atom morphologies, fine-graining is somewhat trivial, as the hydrogens can be placed around the appropriate beads based on sp hybridization rules and typical element-hydrogen bond-lengths. The molecules are then split into electronically active chromophores, which are defined as individual monomers for simplicity. Although the carrier delocalization length for P3HT is around 7 monomers [37,38], we have found that using individual monomers broadly captures the delocalization behaviour by calculating fast electronic couplings between adjacent monomers compared to adjacent chains. Using single-monomer chromophores is advantageous as it removes the requirement of knowing the delocalization length of the simulated polymer beforehand, increasing the transferability and applicability of the model to other polymeric systems [29]. The neighbors of every chromophore in the system are calculated by performing a Voronoi analysis that treats adjacent Voronoi cells as direct neighbors. The molecular orbital energy levels of each pair of neighboring chromophores as a dimer, as well as each chromophore in isolation, are calculated using fast, semi-empirical QCCs. MorphCT uses the Zerner’s intermediate neglect of differential overlap method (ZINDO/S), which has been shown to provide good agreement of relative orbital energies when compared to more rigorous DFT techniques (See Appendix A for details). The energy splitting in dimer method is then used to calculate the electronic coupling between chromophore${}_{i}$ and chromophore${}_{j}$, $J_{i,j}$ [33,39,40]:(1)$$J_{i,j}=\frac{1}{2}\sqrt{{\left(E_{\mathrm{HOMO}}-E_{\mathrm{HOMO}-1}\right)}^{2}-{\left(\Delta E_{i,j}\right)}^{2}},$$
where $E_{\mathrm{HOMO}}$ and $E_{\mathrm{HOMO}-1}$ are the highest and second-highest occupied molecular orbitals of the dimer pair respectively and $\Delta E_{i,j}$ is the site-to-site energy difference, which is calculated as $E_{\mathrm{HOMO},j}-E_{\mathrm{HOMO},i}$. The rate at which a hole is able to hop from *i* to *j* is given by an expression based on semi-classical Marcus theory [41]:(2)$$k_{i,j}=\frac{|J_{i,j}{|}^{2}}{\hbar }\sqrt{\frac{\pi }{\lambda k_{B}T}}\exp\left(-\frac{r_{i,j}}{\alpha }\right)\exp\left(-\frac{{\left(\Delta E_{i,j}-\lambda \right)}^{2}}{4\lambda k_{B}T_{\mathrm{KMC}}}\right),$$
where *ℏ* is the reduced Planck’s constant, λ is the reorganization energy, $k_{B}$ is the Boltzmann constant, and $T_{\mathrm{KMC}}$ is the temperature of the KMC simulation. This temperature, $T_{\mathrm{KMC}}$ is not to be confused with the temperature of the MD simulation, *T*. In this case, $T_{\mathrm{KMC}}$ corresponds to the temperature at which the performance of the device is tested, after processing when the structure has already been “locked in“. For simplicity, we keep this constant at room temperature ($T_{\mathrm{KMC}}=293$ K). We also include an additional exponential term in the hopping rate equation based on the center-of-mass separation between chromophores, $r_{i,j}$, and a tuning parameter α=2 Å. This term originates from Mott’s variable range hopping theory (VRH) [42], which is often used in polymer hopping theory [43,44]. VRH accounts for deficiencies in the prediction of transfer integrals within amorphous systems using the above method, which do not sufficiently suppress the electronic coupling between chromophores with large separations, leading to unphysical carrier motion [31]. The reorganization energy, λ, is the energy required to polarize and depolarize a chromophore, in response to a carrier hopping from one to another. This is material specific, and for P3HT we set λ=306 meV based on electronic structure calculations for a single monomer [45].

After the hopping rates have been calculated between every pair of neighboring chromophores in the system, KMC simulations are performed to explore the motion of carriers throughout the morphology. KMC is an event-based technique, which uses a uniformly generated random number in the interval (0, 1), *x*, to calculate the wait time until a particular event, τ, based on the corresponding rate coefficient, *k*:(3)$$\tau =\frac{-\ln x}{k}.$$

The KMC algorithm then queues up the wait times and selects the event with the shortest wait time as the next event to occur chronologically, implicitly assuming it to be the most probable event. The wait time is then subtracted from all other events in the queue and any new wait times are queued up. MorphCT can consider several types of events that are relevant for organic electronic device operation, such as photo- and dark-injection, as well as exciton dynamics. However, in this work, only one event type is considered—the hop of a charge carrier from chromophore${}_{i}$ to chromophore${}_{j}$ with a rate calculated using Equation (Equation 2).

By repeating this algorithm hundreds of thousands or millions of times, a carrier’s trajectory through the system can be mapped and its total displacement calculated. The morphologies in this investigation are periodically bound, allowing carriers to move hundreds of nanometers through a periodically repeating system comprised of ∼10 nm unit cells. We calculate the mean squared displacement (MSD), averaged over 10,000 carriers each initialized on a randomly selected chromophore, for seven different total simulation times, *t*. The gradient of the MSD as a function of *t* gives the carrier diffusivity, *D*: (4)$$D=\frac{1}{2n}\cdot \frac{d\mathrm{MSD}}{dt},$$
where n=3 is the number of dimensions. *D* can be related to the mobility, μ, through the three-dimensional Einstein-Smoluchowski relation:(5)$$\mu =\frac{qD}{k_{B}T_{\mathrm{KMC}}},$$
where *q* is the unit charge. The relation in Equation (Equation 4) is frequently employed in charge transport investigations [39,46,47], and is expected to provide a reasonable upper-bound for carrier diffusivity for systems with no external driving force [48]. Since our carrier trajectories are obtained in isolation (i.e., no Coulombic interactions with other carriers) and no external electric field is applied, we therefore expect the result to be a “best case” zero-field carrier mobility, $\mu_{0}$, that describes the diffusion of the carriers at low charge density, similar to time-of-flight experiments. Given the absence of other charges in the system restricting carrier movement, and the small, periodically repeating simulation volume enhancing the effect of order in the system, we expect our mobility values to be somewhat larger than those determined experimentally. However, we argue that the simulated mobilities calculated in this work are still an important proxy for semi-conducting electronic device performance, and can be compared to each other to make predictions about expected charge transport trends in physical devices.

## 3. Results and Discussion

Here we calculate the zero-field hole mobilities in P3HT morphologies predicted with molecular simulations in [12]. With both morphologies and predicted mobilities in-hand, we first evaluate structural correlations with mobility by comparing two order parameters. Second, we perform simulations of polydisperse P3HT chains to investigate a mobility anomaly observed for semi-ordered monodisperse chains.

### 3.1. Structure and Mobility in “Small” Morphologies

In previous work, we predicted equilibrium morphologies of P3HT at ∼100 combinations of temperature, *T*, and solvent strength, $\varepsilon_{s}$ [12]. Each of these model systems is monodisperse, with 100 15-mers in a cubic periodic volume with 7 nm edges. At each (*T*, $\varepsilon_{s}$) state-point, we calculate the degree of order, ψ, in the system by identifying clusters of π-stacked thiophene rings with close positions and orientations. The resultant phase diagram is shown in Figure 2a. A band of highly ordered morphologies is visible spanning $\varepsilon_{s}$ from low T∼300 K and good solvent $\varepsilon_{s}=0.2$ to high T∼700 K and poor solvent $\varepsilon_{s}=1.2$. This band is surrounded by state-points at T<300 K and T>1000 K that show a poor degree of order regardless of the solvent quality.

The zero-field hole mobilities, as calculated by MorphCT for each morphology state-point (∼100 systems), are shown in Figure 2b. These mobilities span an order of magnitude from $0.01\leq \mu_{0}<0.15$ cm${}^{2}$/Vs as *T* and $\varepsilon_{s}$. We note these mobilities are roughly two orders of magnitude higher than observed in experiment ($\mu =1\times 10^{-5}$ to $1\times 10^{-3}$ cm${}^{2}$/Vs for time-of-flight measurements [18,19,20,21]), but are still an improvement over similar P3HT calculations ($\mu =1\times 10^{0}$ to $1\times 10^{3}$ cm${}^{2}$/Vs [28,31,32]). First and foremost, we attribute the overprediction to the periodic volume with only 7 nm edges; there is little opportunity for boundaries between transport domains to emerge. That is, periodic volumes may overpredict mobility because grain boundaries (or their analogues) are rare. Second, contaminants such as residual solvent are not represented in our molecular model, and would otherwise lead to restricted mobility in experiments. The lowest $\mu_{0}$ are seen at the highest processing temperatures (>1100 K) in poor solvents ($\varepsilon_{s}$ = 1.2). Conversely, the highest $\mu_{0}$ values are seen in the morphologies prepared at low temperatures (<200 K) in good solvents ($\varepsilon_{s}≃$ 0.2). A band of high mobility is seen in a qualitatively similar region to Figure 2a, spanning from T∼300 K and $\varepsilon_{s}=0.2$ to T∼700 K and $\varepsilon_{s}=1.2$. This suggests that the strong ordering of thiophene backbones into large crystalline clusters is an important prerequisite for efficient carrier transport. However, it is clear that this is not the only factor affecting the transport, as state-points with low T<300 K and $\varepsilon_{s}<0.5$ are also shown to have high mobilities, despite a reduced ψ value. We therefore deduce that our simple definition for ψ, which only considers the conjugation and crystallization of the thiophene backbones, does not sufficiently encode all of the morphological features required to describe charge transport in the system.

In an attempt to better correlate nanostructure to mobility, we propose a new order parameter that supplements ψ with additional structural information not otherwise encoded by the calculation of structural order. Here we utilize disorder in aliphatic side-chains: it is straightforward to calculate bond-stretching statistics, these are structural metrics not included in the clustering criteria of ψ, and it is plausible that high variance in bond-stretching is a proxy for disorder between the thiophene rings that leads to lower mobility. Note that only the bonds in the aliphatic sidechains are considered in this way, as the backbone thiophene rings are rigid and so have fixed bond, angle, and dihedral constraints. We define $\tilde{\sigma_{i}}$ for the morphology at each unique state-point (subscript “*i*”), which is the standard deviation of the bond length distribution for the state-point, $\sigma_{i}$, normalized by the minimum value of $\sigma_{i}$ across all state-points:(6)$$\tilde{\sigma_{i}}=\frac{\sigma_{i}}{\min\left\{\sigma_{i}\right\}}.$$

Our new order parameter $\psi^{\prime }$ is defined by:(7)$$\psi_{i}^{\prime }=\frac{\psi_{i}}{\tilde{\sigma_{i}}}.$$

The aliphatic side-chain bond-length distribution is therefore being used as a proxy for disorder within each cluster, with $\psi^{\prime }$ weighting clusters with a narrow bond-length distribution as more highly ordered than those with broader distributions. This normalization of ψ provides a new lens for structure, as shown in Figure 2c. Qualitatively, $\psi^{\prime }$ better matches the mobility trends in Figure 2b than Figure 2a. This agreement suggests that $\psi^{\prime }$ alleviates shortcomings in ψ (a binary classifier that considers two molecules as only clustered or not), by instead allowing us to quantify *how ordered* a cluster is. We note that $\psi^{\prime }$ still tends to underpredict $\mu_{0}$ at high temperatures >750 K, especially for systems dissolved in poor solvents with $\varepsilon_{s}\leq 0.6$. However, for experimentally relevant temperatures and good solvents, there is broad agreement between $\psi^{\prime }$ and $\mu_{0}$, which is encouraging for the development of a structural metric that can predict charge transport properties of a morphology without performing KMC. The correlation between $\psi^{\prime }$ and $\mu_{0}$ is quantified in Figure 3. The two properties are shown to track better than random, with a correlation coefficient $R^{2}=0.62$ across all state-points, although we note increased variability in $\mu_{0}$ for systems with low $\psi^{\prime }$. Many of the low $\psi^{\prime }$ values correspond to systems at high temperatures (>750 K), suggesting that $\psi^{\prime }$ tends to overpredict the backbone disorder arising from thermal contributions. Generally however, correlation between $\psi^{\prime }$ and $\mu_{0}$ suggests that $\psi^{\prime }$ can be used to quickly identify processing protocols that are expected to have good charge transport properties to submit for further investigation.

### 3.2. Structure and Mobility in “Large” and Polydisperse Cases

To investigate our hypothesis that the high charge mobilities predicted above in “small” volumes is a consequence of these volume we perform charge mobility calculations on systems with 10 times as many simulation elements. Although they contain 1000 oligomers, each edge of these volumes is just over twice as long (15 nm) as the 100 oligomer simulations (7 nm) because the cubic box length scales as the cube root of the number of elements. These volumes are expected to be larger than the average size of the P3HT crystallites, allowing for multiple crystalline and amorphous domains in the same sample in accordance with experiment, but are still smaller than the thickness of the thin-films developed experimentally. While these larger systems hold promise for giving better insight into charge transfer pathways, we recognize the periodic volumes could still lead to inflated absolute mobilities compared to time-of-flight measurements.

#### 3.2.1. Mobility and Carrier Behavior

The mobilities $\mu_{0}$ for the three classes of monodisperse 1000-molecule P3HT morphologies: amorphous ($\psi^{\prime }∼0.17$), semi-crystalline ($\psi^{\prime }∼0.25$), and crystalline ($\psi^{\prime }∼0.33$) are shown in Figure 4a, along with error bars representing the standard deviations from 10 independent microstates. Additionally, all calculated charge transport parameters, along with their associated standard errors, are listed in Table 1. Unlike in the “small” systems, we observe no correlation between mobility and $\psi^{\prime }$. The amorphous and crystalline cases have mobilities commensurate with the “small” systems (Figure 2), whereas the semi-crystalline system exhibits a significantly lower mobility. We can interpret this observation in two opposing ways: On one hand, zero-field charge mobility of $\mu_{0}=1.56\times 10^{-2}$ cm${}^{2}$/Vs is nearing the $1\times 10^{-3}$ cm${}^{2}$/Vs observed in experiments, and seems to support our working hypothesis that boundaries between crystallites should inhibit charge transport. On the other hand, this observation is surprising because P3HT is widely regarded to form a semi-crystalline structure in experimental devices, which we expect to have higher charge mobility than the amorphous case [49]. We note that further modifications to $\psi^{\prime }$ that explicitly accounts for the variation in transfer integrals across chromophores does not address the fact that the lowest mobility comes from medium order (See Appendix A for details). Throughout this work (e.g., Figure 4) we color backbones of P3HT based upon the cluster to which they belong, which depends on charge hops, and is discussed in detail in Appendix A.

To further investigate the anomalous semi-crystalline case, we consider the directions charges move during the KMC simulations. It might be expected that the carrier trajectory anisotropy controls the overall mobility—a high anisotropy suggests that carriers are restricted to a particular direction, making it more likely to increase its mean squared displacement over the same amount of time than in a system where transport in three dimensions is equally likely. In Figure 4b, carrier transport is shown to be anisotropic in the crystalline morphology, indicating a consistent grain orientation between the crystalline regions. The anisotropy is significantly lower in the semi-crystalline case, where a variety of grain orientations are present. Perhaps unsurprisingly, the amorphous systems exhibit near-spherical carrier transport, which is consistent with the lack of ordered crystallines in the morphologies. The anisotropy is somewhat higher in the semi-crystalline case, where a variety of grain orientations are present. In Figure 4b, carrier transport is shown to be anisotropic in the crystalline morphology, indicating a consistent grain orientation between the crystalline regions. Given that the anisotropy of the semi-crystalline morphology is intermediate between the more and less ordered systems (unlike the calculated mobilities), we deduce that anisotropy is not the sole factor governing carrier mobility.

Our calculated hopping rate distributions presented in Figure 4c–e in isolation would also suggest intermediate mobilities for the semi-crystalline system, as the availability of fast (high $k_{i,j}∼10^{14}$ s${}^{-1}$) inter-molecular hops appears to decrease with decreasing $\psi^{\prime }$. Therefore, the distribution of hopping rates alone is insufficient to predict performance—the rate, location, and neighborhood of those hops in the morphology are all required in order to make predictions.

The amorphous morphology (Figure 4f) explicitly shows no crystallization, instead forming a disordered matrix of entangled polymer chains. However, with the chains colored based on charge hopping, it is clear that there is one large (red) percolating cluster that connects most chains to most other. The structure of the crystalline morphology (Figure 4h) is expectedly lamellar, with one large, ordered percolating (red) cluster. The semi-crystalline system (Figure 4g) shows small regions of crystallized lamellae, interspersed within an amorphous matrix. The prevalence of multiple clusters indicates that charges have trouble hopping between crystallites. This is the first evidence we observe of non-intermediate properties of the semi-crystalline system compared to the crystalline and amorphous morphologies. Further analysis of the hops occurring within the ordered crystallites of the semi-crystalline and crystalline morphologies reveals that charges can travel in fast “loops” within ordered regions wherein hops are fast, but total displacement is low (Appendix A). That is, carriers in ordered regions have a high probability of spending long periods of time hopping between the same set of chromophores within the same plane, without increasing their displacement from their start position. Because these carriers are effectively “trapped”, if the ordered regions are not connected, overall mobility suffers.

Considering these factors in aggregate, we conclude that the crystalline morphology mobility of 1.16 $\times 10^{-1}$ cm${}^{2}$/Vs is due primarily to fast carrier transport along the ordered crystallites, and note that it would be even higher if carriers did not frequently “loop around” within the large ordered regions rather than travel ballistically. The mobility through the amorphous morphology is restricted by slower overall isotropic carrier motion, but the lack of traps explains higher mobility than the semi-crystalline case. The proximity of the amorphous case mobility ($1.02\times 10^{-1}$ cm${}^{2}$/Vs) to the crystalline case highlights the importance of trapping to overall carrier mobility. This also bears out some recent investigations that have shown beneficial carrier behavior in less conventionally-ordered systems [50]. The semi-crystalline morphology ranks highly in isotropic transport, *and* low in ordered domain alignment, resulting in an order-of-magnitude lower mobility of $1.64\times 10^{-2}$ cm${}^{2}$/Vs.

In summary, we find that a convolution of different structural and transport metrics is required in order to correctly predict carrier mobilities—no one factor is sufficient to explain the observed trend. Carrier transport is strongly dependent on the local neighborhood around each chromophore—if a carrier has easy access to the surrounding chromophores but not beyond, then it will become trapped, even if the average cluster and chromophore characteristics of the whole morphology are favourable. KMC simulations are the current best way to convolve the structural metrics and obtain the device performance behavior—it is presently not possible to map directly between structure and performance otherwise. For the three degrees of order considered here, we have shown that the amorphous morphology has stronger connections (characterized by a smaller number of larger clusters) than the more-ordered semi-crystalline one, leading to a higher carrier mobility and improved charge transport. That this disordered charge mobility is higher than expected in experiments suggests there are improvements to the absolute value charge hopping rates, or assumptions about chromophore size and electron delocalization that could improve mobilities calculated with KMC.

#### 3.2.2. Tie Chains in Polydisperse Systems

We hypothesize that the difference in mobility between the semi-crystalline system and the crystalline and amorphous morphologies is due to the monodispersity and short length of the chains studied here. Generally in experimental devices, P3HT is obtained with a molecular weight in excess of 50 kDa, corresponding to chain polymerisation lengths of many hundreds or thousands of monomers [28]. In such systems, the chains are long enough to fold back on themselves several times, forming sheet-style crystallites in the system, where a single chain can form multiple layers of the same crystallite [51]. Previous work has shown that, while 15mer chains were able to reproduce experimental scattering patterns, they were too short to undergo self-folding. Conversely, chains with 50 repeat units were able to undergo self-folding, but required untenably long simulation times to order into the experimentally observed structures [12]. In the case of longer chains, multiple folds and multiple chains can stack together to increase the size of the crystallite regions, with portions of the outermost chains remaining outside of the crystallite, forming an amorphous matrix between the grains. An example of this is seen in Figure 5a,b in which a series of longer chains or a single long chains spans multiple clusters. In some cases, these “tails” may connect to a different crystallite, effectively forming a “tie-chain” between two crystallites [49]. Since carrier motion is faster along a chain than between neighboring chains, this provides a fast and efficient route for carriers to transport between crystallites, which is sometimes known as a “carrier highway”. Such routes are not unique to P3HT; tie-chains are found to be critical in other polymeric systems, for instance in complimentary semiconducting polymer blends [52]. In our work, chains with 15 repeat units do not self-assemble tie-chains as evidenced by the semi-crystalline system explored here.

To test whether longer chains will serve as tie-chains between otherwise charge-trapping crystallites, we investigate morphologies made with polydisperse chains up to 50 repeat units (∼8 kDa). The maximum of 50 repeat units is chosen to prevent any individual chain from interacting with itself in more than one image of the periodic volume. Due to these size constraints, we are unable to achieve experimentally relevant chain lengths (for example, 20–100 kDa from Sigma-Aldrich, St. Louis, MO, USA or Reike Metals, Lincoln, NE, USA). We can, however, achieve experimental polydispersities of 1.8 (See Appendix A for details). Furthermore, simulating a polydisperse distribution of chain lengths allows us to introduce chains that may be long enough to span several crystallites, while still maintaining appropriate length-scales to obtain good agreement with experimental scattering patterns (also demonstrated in Appendix A).

Polydisperse morphologies are generated using the same process as the monodisperse cases as explained previously, and result in three similar degrees of ordering: amorphous ($\psi^{\prime }∼$ 0.18), semi-crystalline ($\psi^{\prime }∼$ 0.27), and crystalline ($\psi^{\prime }∼$ 0.31). We calculate mobilities of these polydisperse morphologies with KMC and present them in Figure 6a. By including a distribution of chain lengths, the expected order-mobility trend has been reclaimed—mobility increases with additional order. Generally, $\mu_{0}$ is slightly higher in the polydisperse systems than in the monodisperse 15mer systems, as the increased average molecular weight (2.9 ± 0.1 kDa for the polydisperse and 2.5 kDa for the monodisperse systems) leads to a higher proportion of fast intra-chain hops. Figure 6b–d show that, unlike the monodisperse systems in Figure 4f–h, all three of the systems are highly connected and form a single, large cluster spanning the entire system (colored red). This higher connectivity is due to the presence of more chains spanning between crystallites in the polydisperse case than the short monodisperse case (Figure 5c). The improved connectivity is quantified in Table 2, where the number of large clusters and the size of the largest cluster are both intermediate between the amorphous and crystalline systems. Additionally, Table 2 shows a significantly lower carrier trajectory anisotropy in the case of the semi-crystalline and crystalline polydisperse systems than in the monodisperse case (Table 1). This suggests that charges are no longer restricted by grain boundaries and are able to change direction more easily—a process that was prohibitively slow in the monodisperse case. These results are in good agreement with previous investigations that show tie-chains are a dominating factor in carrier transport through polymer devices [28,53].

The observation that $\psi^{\prime }$ and $\mu_{0}$ are strongly correlated in large, polydisperse systems (Figure 6), somewhat correlated in small, monodisperse systems (Figure 3), and poorly correlated in large, monodisperse systems (Figure 4a), highlights a shortcoming in using purely structural metrics to predict charge transport. In isolation, structure can provide some insight into the average rate at which hops can occur in the morphology—of the hopping criteria studied in this investigation, only the hopping rate is described by $\psi^{\prime }$. This relationship is quantified by the increase of average inter-molecular hopping rates for both the monodisperse (Table 1 shows $0.834\to 2.208\to 2.642\times 10^{13}$ for the amorphous, semi-crystalline, and crystalline structures respectively) and polydisperse systems (Table 2: $0.700\to 1.231\to 1.590\times 10^{13}$). Graphically this is also demonstrated by the shift of the inter-molecular hopping rate peak towards the intra-molecular peak in Figure 4c–e. However, considering only the hopping rate distributions fails to take into account the local neighborhood of hops available. Therefore, $\psi^{\prime }$ is unable to distinguish between regions where charges may be trapped within crystallites, or able to flow along a fast extended path. This is confirmed by our clustering analysis in Appendix A—no combination of purely structural cluster criteria was able to produce the same cluster distributions observed in our simulations. We therefore conclude that knowledge of the carrier hopping rates in the chromophore network is insufficient—one must also know how these rates are distributed in order to identify regions of trapping that will reduce carrier mobilities. This is a key advantage of computational methods such as KMC—even though carriers have no knowledge of the surrounding hop neighborhood (all hops are performed on a chromophore-by-chromophore basis to first order), the extensive statistical averaging of the method allows us to probe the local hopping neighborhood and identify crystallites.

## 4. Conclusions

Using QCC to inform KMC simulations of charge transport in P3HT morphologies currently gives the best insight into how nanostructure influences charge mobility. These calculations confirm that charges move most quickly along P3HT backbones and second-most quickly between aligned backbones. However, because charges rarely hop between distinct crystallites, tie-chains connecting ordered crystallites are essential to mitigating the trapping of charges that would otherwise lower mobility. By combining the large volumes from optimized MD simulations of P3HT with QCC-informed charge transport, this is the first work to definitively show the impact tie-chains have on charge mobility. The computational techniques demonstrated in this manuscript are applicable to other organic semiconducting materials (including non-polymeric small molecules) and we expect to detect a similar relationship between charge transport and the presence of tie-chains for other conjugated polymer systems.

Looking to the future, this work highlights two areas for improving mobility predictions. Firstly, the present work shows that purely structural metrics miss important factors for charge transport, but this does not preclude the existence of better metrics that are more predictive than those studied here. That is, discovery of structural metrics that are good enough to predict mobility without having to perform KMC simulations would save a lot of time. Secondly, while the mobilities predicted with KMC are the current state-of-the-art, they are systematically about two orders of magnitude higher than in experiments. Whether this is due to inaccurate assumptions about what comprises a chromophore, or whether improvements to calculating charge hopping rates are needed, or something else, it seems like quantitative predictions of mobility are on the horizon. Exploring these improvements to the KMC calculations presented here and investigating a broader range of chemistries to further validate these techniques is the subject of future work.

## Figures and Tables

**Figure 1 polymers-10-01358-f001:**
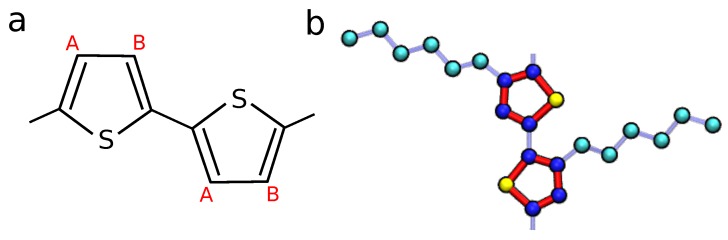
(**a**) The chemical structure of a poly-alkylthiophene chain. If the aliphatic side chains are consistently located only at the A sites or the B sites along the backbone, then the polymer is regioregular. In regiorandom chains, the placement of the aliphatic side chain is randomly placed at an A or B site on each thiophene; (**b**) P3HT united-atom model used in previous work. Sulfur, aromatic carbons, and aliphatic carbons are represented by yellow, dark blue, and cyan beads respectively. The rigid bonds are shown with thick red lines and flexible bonds are shown in light blue. (Figure replicated from [12] with authors’ permission).

**Figure 2 polymers-10-01358-f002:**
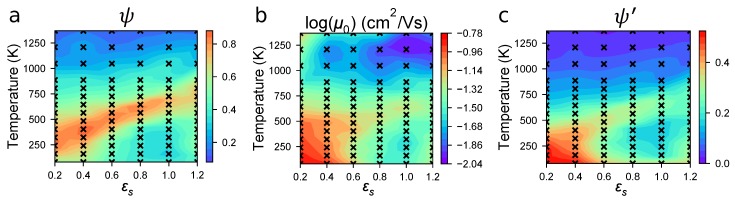
Heatmaps of the various properties explored for each simulation. In all cases, black “x”s correspond to state-points where simulations were conducted, with values in between determined by linear interpolation. Red regions correspond to a large value of the property, whereas blue regions correspond to a smaller value. Color bars are normalized to the maximum value of each parameter. (**a**) The structural order of each system given by the order parameter, ψ, as in the previous work (Figure replicated from [12] with authors’ permission); (**b**) The hole mobility, $\mu_{0}$, varying between red (∼0.15 cm${}^{2}$/Vs) and blue (∼0.01 cm${}^{2}$/Vs) There is not a perfect mapping between ψ and $\mu_{0}$—lower and higher temperature systems have higher and lower $\mu_{0}$ respectively, which is not captured effectively by ψ; (**c**) The modified order parameter, $\psi^{\prime }$, created by normalizing ψ by the standard deviation of aliphatic bond lengths. $\psi^{\prime }$ is a significant improvement over ψ, as it captures the presence of additional disorder in the system.

**Figure 3 polymers-10-01358-f003:**
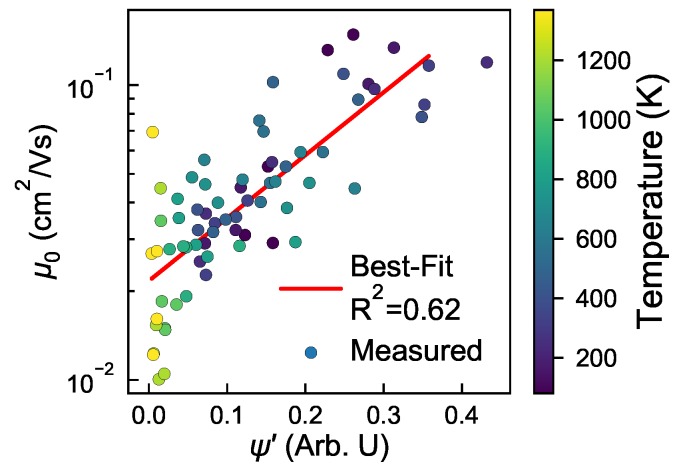
The zero-field mobility, $\mu_{0}$ shows reasonable correlation to $\psi^{\prime }$ ($R^{2}$ = 0.62), indicating it can be useful as a purely structural metric to broadly predict interesting processing state-points to investigate further.

**Figure 4 polymers-10-01358-f004:**
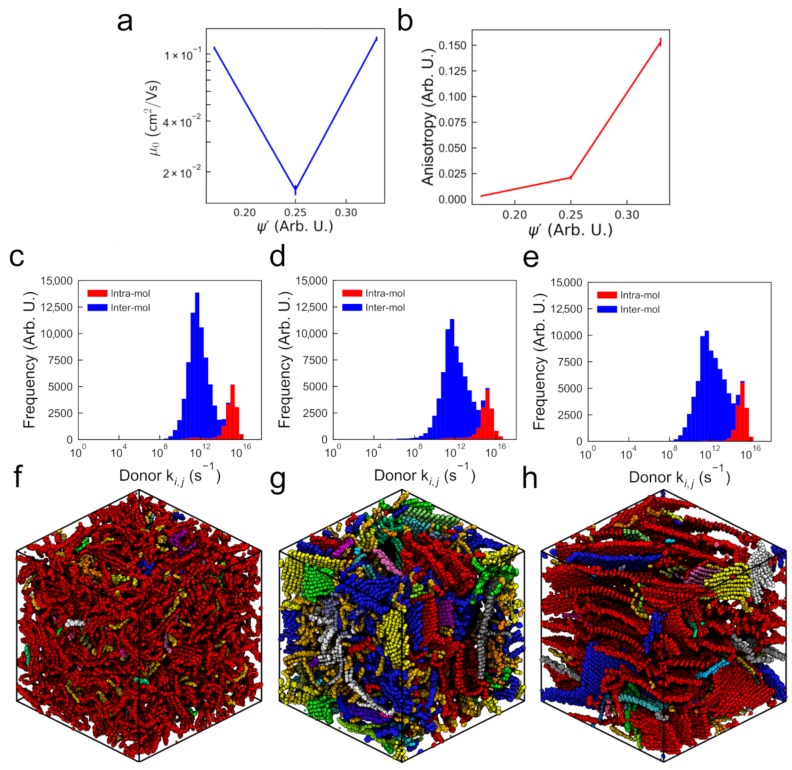
(**a**) Zero-field hole mobility (inset: representative morphology visualization with sidechains omitted for clarity) and (**b**) carrier trajectory anisotropy for the 1000 oligomer simulations. In both cases, error bars are calculated based on the standard error arising from 10 repeat simulations of structurally decorrelated morphologies simulated under the same conditions. The hopping-rate distributions for the (**c**) amorphous, (**d**) semi-crystalline, and (**e**) crystalline systems have stacked bars (no obfuscation) where red describes hops along the chain and blue describes hops between chains. Regions of high connectivity in the (**f**) amorphous, (**g**) semi-crystalline, and (**h**) crystalline morphologies are denoted by colored clusters. The clusters are defined based on the frequency of hops performed between chromophore pairs in the simulations (further details in Appendix A).

**Figure 5 polymers-10-01358-f005:**
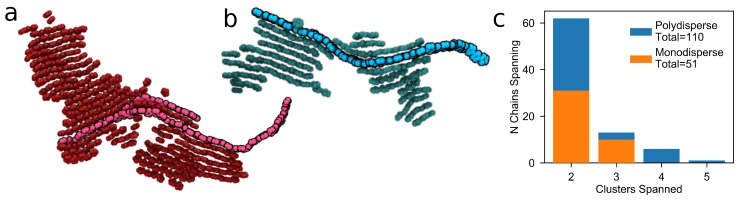
Long polymer chains are able to link clusters together to enhance charge transport between them. The links can either consist of (**a**) multiple chains or (**b**) a single chain extending through the surrounding amorphous matrix; (**c**) The semi-crystalline polydisperse systems, with chains up to 50 monomers and polydispersities of 1.8, have double the amount of tie-chains spanning two clusters as the semi-crystalline monodisperse system consisting only 15mers. Additionally, some chains in the polydisperse system span four or five clusters. The bars in the histogram overlap so that the frequency of chains spanning multiple clusters is given by the top of the orange and blue bars for the mono- and polydisperse systems respectively.

**Figure 6 polymers-10-01358-f006:**
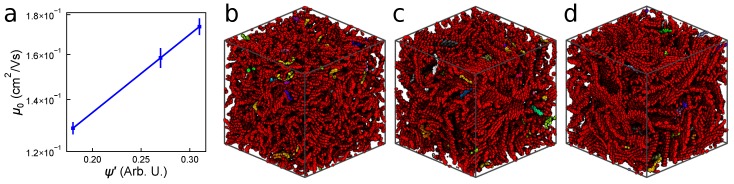
(**a**) Zero-field mobilities for the polydisperse P3HT simulations based on the modified order parameter $\psi^{\prime }$; Morphologies showing regions of high connectivity for the (**b**) amorphous; (**c**) semi-crystalline; and (**d**) crystalline systems.

**Table 1 polymers-10-01358-t001:** Charge transport metrics calculated for the three degrees of ordering in systems of 1000 monodisperse P3HT chains. Average values for 10 statistically independent samples are listed, along with the corresponding standard error over the 10 measurements. Clusters are defined based on a hopping frequency cut-off as described in the text.

Property	Amorphous	Semi-Crystalline	Crystalline
Mobility (cm${}^{2}$/Vs)	$\left(1.085\pm 0.006\right)\times 10^{-1}$	$\left(0.156\pm 0.003\right)\times 10^{-1}$	$\left(1.23\pm 0.01\right)\times 10^{-1}$
Anisotropy (Arb. U.)	0.0031±0.0001	0.0210±0.0006	0.153±0.001
Intra-molecular rate (s${}^{-1}$)	$\left(1.813\pm 0.001\right)\times 10^{15}$	$\left(2.493\pm 0.001\right)\times 10^{15}$	$\left(1.8703\pm 0.0006\right)\times 10^{15}$
Inter-molecular rate (s${}^{-1}$)	$\left(0.834\pm 0.001\right)\times 10^{13}$	$\left(2.208\pm 0.005\right)\times 10^{13}$	$\left(2.642\pm 0.005\right)\times 10^{13}$
$\Delta E_{ij}$ std (eV)	0.06252±0.00006	0.1114±0.0001	0.0571±0.0001
Total clusters (Arb. U.)	500±10	1540±10	467±6
Large (>6) clusters (Arb. U.)	134±1	209±1	73±1
Largest cluster size (Arb. U.)	9600±200	2100±100	8300±300

**Table 2 polymers-10-01358-t002:** Charge transport metrics calculated for three degrees of order in polydisperse P3HT systems. Average values over 10 statistically independent samples are listed, along with the corresponding standard error over the 10 measurements. Clusters are defined based on a hopping frequency cut-off as described in the text.

Property	Amorphous	Semi-Crystalline	Crystalline
Mobility (cm${}^{2}$/Vs)	$\left(1.29\pm 0.02\right)\times 10^{-1}$	$\left(1.58\pm 0.04\right)\times 10^{-1}$	$\left(1.74\pm 0.04\right)\times 10^{-1}$
Anisotropy (Arb. U.)	0.0040±0.0005	0.016±0.001	0.020±0.004
Intra-molecular rate (s${}^{-1}$)	$\left(1.3413\pm 0.0003\right)\times 10^{15}$	$\left(1.4670\pm 0.0007\right)\times 10^{15}$	$\left(1.5137\pm 0.0002\right)\times 10^{15}$
Inter-molecular rate (s${}^{-1}$)	$\left(0.700\pm 0.004\right)\times 10^{13}$	$\left(1.231\pm 0.0007\right)\times 10^{13}$	$\left(1.590\pm 0.007\right)\times 10^{13}$
$\Delta E_{ij}$ std (eV)	0.0554±0.0002	0.0549±0.0001	0.0538±0.0001
Total clusters (Arb. U.)	400±23	350±17	380±21
Large (>6) clusters (Arb. U.)	130±10	70±6	60±3
Largest cluster size (Arb. U.)	11,200 ± 260	13,200 ± 200	13,500 ± 100

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
