# Peer review of "Tying Together Multiscale Calculations for Charge Transport in P3HT: Structural Descriptors, Morphology, and Tie-Chains"

_polymers, 2018, doi:10.3390/polym10121358_

Round 1

Reviewer 1 Report

Comments on “Tying together multiscale calculations for charge transport in P3HT: structural descriptors, morphology, and tie-chains” by Evan Miller et al.

In this work, the authors have combined the quantum chemical calculations (QCC) and kinetic Monte Carlo (KMC) simulations to predict the zero-field hole mobilities of poly(3-hexylthiophene) polymer, P3HT. The different simulation volume, structure order, chain length polydispersity and morphology of P3HT polymers are considered. The P3HT polymers with different morphologies are generate by using the  optimized molecular dynamics force-field, which can represent a spectrum of nanostructured order. The hole mobility of P3HT polymer is found to be correlated with the backbone clustering and disorder from side-chain conformations. In addition, the efficient charge transport requires the strongly interconnected thiophene backbones, indicating the important role played by the “tie-chains” in mobile changes of P3HT polymers. Overall, the manuscript is well-written and scientifically sounds. The manuscript can be accepted after considering the following minor suggestions.

1)      The zero-field mobilities have been computed. How these values compare with existing experimental and simulation results?

2)      The amorphous, semi-crystalline and crystalline P3HT polymers are considered. However, it is not clear how the authors generate these models and define the crystallinities in these models.

Author Response

We thank the reviewers for taking the time to review our manuscript and we include our responses in the attached pdf.

Reviewer 2 Report

In this article, the authors introduced a new method to calculate the mobility of charge carrier in P3HT and reported the relationship the mobility and the morphology of condensed P3HT.

   I expect that the calculation method presented in this article gives various helpful information to develop next-generation organic opto-electric devices.  However, there are some unclear arguments due to the empirical approach in the calculation method. 

(1) In p.5, the value of T_KMC is chosen to be 293K.   Why did the authors choose that T_KMC=293K?

(2) In p.5, the authors used the relationship between the diffusivity and the mobility shown by eq.(4) on the basis of the fluctuation-dissipation theorem.  In general, the fluctuation-dissipation theorem is valid for large particle moving slowly in medium composed of small particles moving quickly such as colloidal particle.  I don’t think the fluctuation-dissipation theorem is suitable for the charge carrier because the carrier particle is quite small and there are no particles much smaller than the carrier particle surrounding the carrier particle. Why did the authors use eq.(4) and why did the authors choose the temperature in eq.(4) to be T_KMC.

(3) In p.7, what does the new order parameter \psi’ mean?  Please explain the difference between \psi and \psi’ based on the morphology.  Please show the “definition” of \psi again in this article.

(4) In 3.2.1, three types of morphologies, amorphous, semi-crystalline and crystalline are introduced.  How did authors determine the type of morphology?  Is it based on the value of \psi’?

(5) Please write units of the quantities in Tables 1 and 2 to improve readability.

Author Response

(The authors gave the same response as above.)

Reviewer 3 Report

The authors demonstrated quantum chemical calculations and kinetic Monte Carlo simulations to predict the zero-field hole mobilities of ∼ 100 morphologies of P3HT, with varying simulation volume, structural order, and chain-length polydispersity. This work is very interesting and provides insight to probe the relationship between molecular nanostructure and device performance. For these reasons, this paper can be accepted after solving below issues.

1. This work is focused on the typical donor polymer P3HT, how about the other materials? The authors are suggested to give some discussions.

2. Related papers about charge transport in organic materials should be cited (Adv. Funct. Mater. 2016, 26, 776-783; Light: Sci. Appl. 2016, 5, e16137; Nano-Micro Lett. 2017, 9, 37)

Author Response

(The authors gave the same response as above.)
